# Near-term pregnant women in the Dominican Republic experience high rates of Group B *Streptococcus* rectovaginal colonization with virulent strains

Katherine M. Laycock[1]*, Francia Acosta[2], Sandra Valera[3], Ana Villegas[3], Elia Mejia[3], Christian Mateo[3], Rosa Felipe[3], Anabel Fernández[4], Megan Job[5], Sophia Dongas[5], Andrew P. Steenhoff[4,6,7], Adam J. Ratner[5,8], Sarah Geoghegan[9]

1 The Ryan White Center for Pediatric Infectious Disease and Global Health, Department of Pediatrics, Indiana University School of Medicine, Indianapolis, Indiana, United States of America, 2 Niños Primeros en Salud, Consuelo, Dominican Republic, 3 Hospital Materno Infantil San Lorenzo de Los Mina, Santo Domingo, Dominican Republic, 4 Global Health Center, Children's Hospital of Philadelphia, Philadelphia, Pennsylvania, United States of America, 5 Department of Pediatrics, Grossman School of Medicine, New York University, New York, New York, United States of America, 6 Division of Infectious Diseases, Children's Hospital of Philadelphia, Philadelphia, Pennsylvania, United States of America, 7 Department of Pediatrics, Perelman School of Medicine, University of Pennsylvania, Philadelphia, Pennsylvania, United States of America, 8 Department of Microbiology, Grossman School of Medicine, New York University, New York, New York, United States of America, 9 Division of Paediatric Infectious Diseases, Children's Health Ireland at Crumlin, Dublin, Ireland

* kalayco@iu.edu

**Data Availability Statement:** Whole-genome sequencing data are publicly available in the Sequence Read Archive of the National Center for

## Abstract

Maternal colonization with Group B *Streptococcus* (GBS) is an important cause of stillbirth, prematurity, and serious infection and death in infants worldwide. Resource constraints limit prevention strategies in many regions. Maternal GBS vaccines in development could be a more accessible prevention strategy, but data on geographic variations in GBS clones are needed to guide development of a broadly effective vaccine. In the Dominican Republic (DR), limited data suggest that pregnant women experience GBS colonization at rates among the highest globally. We aimed to determine the prevalence of maternal rectovaginal GBS colonization and describe clonal characteristics of colonizing strains in the DR. A cross-sectional study assessed rectovaginal GBS colonization in 350 near-term pregnant women presenting for routine prenatal care at an urban tertiary center in the DR. Rectovaginal samples were tested with chromogenic Strep B Carrot Broth and cultured for confirmatory whole-genome sequencing. In a secondary analysis, participants' demographics and histories were assessed for association with GBS colonization. Rectovaginal GBS colonization occurred in 26.6% of women. Serotypes Ia, Ib, II, III, IV, and V were detected, with no one serotype predominating; serotype III was identified most frequently (21.5%). Virulent and emerging strains were common, including CC17 (15.1%) and ST1010 (17.2%). In this first characterization of maternal GBS serotypes in the DR, we found high rates of rectovaginal colonization including with virulent and emerging GBS strains. The serotypes observed here are all targeted by candidate hexavalent GBS vaccines, suggesting effective protection in the DR.

Biotechnology Information under BioProject ID PRJNA945321. Individual strain MLST and CC data are available in the PubMLST database under isolate IDs 25413-25505. A deidentified dataset of participant demographics and GBS results is available at https://doi.org/10.7910/DVN/RULBKY.

**Funding:** KML and SG were funded by the Children's Hospital of Philadelphia (CHOP) Melissa Ketunuti Endowed Fellowship in Global Health and Infectious Diseases (https://www.chop.edu/centers-programs/global-health-center). SG was funded by a CHOP Global Health Center Pilot Grant (https://www.chop.edu/centers-programs/global-health-center) and a Clinical Research Fellowship, National Children's Research Center, Dublin (D/19/16) (https://www.nationalchildrensresearchcentre.ie). Swabs used in this study were donated by COPAN Italia (https://www.copangroup.com). The funders had no role in study design, data collection and analysis, decision to publish, or preparation of the manuscript.

**Competing interests:** The authors have declared that no competing interests exist.

# Introduction

Group B *Streptococcus* (GBS) is a leading cause of serious infection and death among infants globally [1]. GBS colonizes the rectovaginal area of an estimated 18% of pregnant women worldwide [2]. Vertical transmission of GBS from colonized women to their infants at or near delivery can cause stillbirth, premature birth, and invasive neonatal disease including sepsis, pneumonia, and meningitis [1]. Routine screening for GBS colonization during pregnancy and administration of intrapartum antibiotic prophylaxis (IAP) have reduced the incidence of early-onset neonatal GBS disease in some populations [3, 4]. However, prevention strategies vary by region. In resource-constrained settings, implementation of maternal screening and IAP remains limited, and neonatal GBS disease remains a significant and likely underdiagnosed problem [1, 3, 5, 6]. Barriers to maternal screening and IAP include cost, insufficient laboratory infrastructure, lack of timely access to skilled peripartum care, and home birth practices [3, 5]. Maternal GBS vaccines are currently in Phase II clinical trials [7]. A safe and effective vaccine could be a more accessible preventive strategy, particularly in low- and middle-income countries (LMICs) [5, 8, 9].

GBS strains have different capsular polysaccharides (CPS) that allow classification into serotypes, with ten serotypes currently recognized: Ia, Ib, II, III, IV, V, VI, VII, VIII, and IX. Additional allelic variations also allow classification by multi-locus sequence types (MLST), which are grouped into clonal complexes (CC). Some specific serotypes, sequence types (ST), or CC are associated with increased virulence [1, 10]. Serotypes III and Ia have been identified most often worldwide in studies of maternal colonization and neonatal invasive disease [10]. CPS is a prime target for vaccine candidates [7]. However, geographic variations in serotype distribution and the limited availability of sequencing data in many regions, especially in LMICs, present challenges to the development of a broadly effective vaccine [10, 11]. More comprehensive data on the regional variation in GBS strains are needed to inform next steps in vaccine development and distribution.

The Caribbean region has the highest prevalence of maternal GBS colonization worldwide, with pooled data estimating colonization in 34.7% (95% confidence interval [CI], 29.5%-39.9%) [2]. Reasons for this high prevalence have not been identified. In the Dominican Republic (DR), limited available data suggest that rates of maternal GBS colonization may exceed those in the rest of the Caribbean region. The only previous study of maternal GBS colonization in the DR to date identified GBS in 43.5% of pregnant women [12]. GBS serotypes responsible for colonization during pregnancy in the DR have not been characterized. As in other LMICs, maternal GBS screening and IAP are not routinely implemented in the DR, highlighting the potential impact of an effective maternal GBS vaccine [13]. Here, we aimed to better characterize the rate of GBS colonization and identify the serotypes present in near-term pregnant women in the DR. In a secondary analysis, we examined potential factors associated with colonization.

# Methods

## Ethics statement

Ethical approval was obtained from the institutional review board of the Children's Hospital of Philadelphia (18–015392) and from the DR's national institutional review board, Consejo Nacional de Bioética en Salud (033–2019). Hospital leadership at the study site also provided written informed consent for institutional participation. All participants provided written informed consent in Spanish and had the opportunity to withdraw from the study at any time. Enrollment had no lower age limit as pregnant minors in the DR are considered emancipated and able to provide informed consent.

## Study setting and design

A cross-sectional study was designed to determine the prevalence of rectovaginal GBS colonization and examine factors associated with colonization in near-term pregnant women in the DR. Spanish-speaking pregnant women at 35 weeks or greater gestational age who presented for routine antenatal care at Hospital Materno Infantil San Lorenzo de Los Mina, Santo Domingo, DR, from September 2021 through March 2022 were eligible for inclusion. The study site is a government-funded urban tertiary maternal and pediatric hospital that logs approximately 10,000 births annually among patients across a range of socioeconomic backgrounds.

After informed consent, structured interviews were conducted in Spanish to collect data on each participant's demographics, medical history, and obstetric history. Participants' ages were recorded in pre-defined age bands (≤17 years, 18–24 years, 25–30 years, 31–35 years, 36–40 years, and >40 years). Variables explored included participants' sexual activity, vaginal hygiene practices, antimicrobial use during pregnancy, and history of adverse birth outcomes and neonatal infections with any prior pregnancies. Anterior vaginal and rectal sampling was then performed to determine the presence of GBS.

All participant data were collected by one study investigator and anonymized using a unique identifying number. Other investigators had access only to anonymized data.

## Sample collection and processing

One anterior vaginal, one rectal, and two rectovaginal samples were collected by a study investigator from each participant using flocked swabs in Liquid Amies medium (COPAN Eswab). One rectovaginal swab was tested per manufacturer instructions using Strep B Carrot Broth One-Step (Hardy Diagnostics), a liquid chromogenic medium selective for GBS. Participants with positive results by Carrot Broth were notified via secure messaging so that they could receive appropriate IAP. The remaining swabs and residual Carrot Broths were frozen at -80˚C prior to confirmatory analysis. Samples were plated directly onto CHROMagar StrepB (CHROMagar), a solid chromogenic medium selective for GBS, and incubated in aerobic conditions at 37˚C for 18–24 hours. Colony growth consistent with GBS was subcultured for whole-genome sequencing (WGS) and subsequent MLST and serotype determination.

## GBS WGS and determination of genomic serotype, MLST, and CC

Genomic DNA was purified from GBS strains on a KingFisher Flex automated instrument using the MagMAX Viral/Pathogen Ultra Nucleic Acid Isolation Kit (Applied Biosystems) in 96-well deep-well plates. Initial lysis was performed in Tissue & Cell Lysis Solution (LGC Biosearch Technologies) with a 6-minute bead beating step with 0.1 mm zirconia/silica beads (BioSpec Products). Library preparation (Illumina DNA Prep) and WGS (Illumina NovaSeq 6000, PE 2x150) were performed at the NYU Genome Technology Core. Adapter sequences were removed, and sequenced were trimmed for quality using Trimmomatic 0.36 with a minimum length of 120 bp [14]. Contamination checks were performed with Kraken 2.0.7 and Bracken 2.5.3 [15, 16]. All strains had >88% of reads classified as *Streptococcus agalactiae*/GBS with most non-GBS reads being assigned to closely related streptococci, consistent with either lateral transfer events or ambiguity in assignment based on the presence of closely related genes. We used SRST2 0.2.0 for read mapping and determined genomic serotype using the GBS-SBG database, MLST/CC using *Streptococcus agalactiae* allele sequences and MLST definition files from pubmlst.org, and antimicrobial resistance alleles (macrolide-lincosamide-streptogramin (MLS), tetracycline, and fluoroquinolone resistance determinants as well as mutations in PBP2x) from Metcalf, et al. [17–20]. Phenotypic antimicrobial susceptibility

testing was not performed. Mashtree was used to construct a tree based on kmer distances, which was then visualized in Microreact [21, 22]. Raw sequencing reads are available in the Sequence Read Archive of the National Center for Biotechnology Information under BioProject ID PRJNA945321. Novel alleles were submitted to pubmlst.org, and new allele and MLST designations were assigned as detailed in S1 Table. Individual strain MLST and CC data are available in the PubMLST database under isolate IDs 25413–25505.

## Statistical analysis

Sample size calculations assumed a maximum colonization rate of 35% [2]. Allowing for a margin of error of 5% at 95% confidence, we targeted a sample size of 350 participants.

Anonymized study data were collected and managed using REDCap electronic data capture tools hosted at the Children's Hospital of Philadelphia. Prevalence of rectovaginal GBS colonization was calculated by dividing the number of positive samples (by Carrot Broth, CHROMagar, or WGS) by the total number of samples collected. Intertest reliability between methods of GBS detection was assessed using the Kappa statistic. In a secondary analysis, participant variables were summarized using descriptive statistics and compared to GBS results using chi-square analyses. A p-value $<0.05$ was considered significant. These analyses were performed using STATA software, version 17.1 (StataCorp, College Station, Texas, USA).

## Inclusivity in global research

Additional information regarding the ethical, cultural, and scientific considerations specific to inclusivity in global research is included in the S1 Checklist.

# Results

We approached 1,365 pregnant women for study inclusion between September 2021 and March 2022, and 486 women met eligibility criteria (S1 Fig). Of those eligible, 350 women participated in structured interviews and rectovaginal sample collection (response rate 72.0%).

## Demographic characteristics

Participants' demographic characteristics are shown in Table 1. The 350 participants had a median age of 18–24 years (interquartile range [IQR], 18–30 years). Exact ages were recorded for 181 participants; these women ranged in age from 13 to 45 years (median age, 19 years; IQR, 17–23 years). Most women (75.4%) were born in the Dominican Republic; the remainder were born in Haiti (23.7%) or Venezuela (0.9%). Most (72.3%) were in a common law marriage, and most (52.6%) had completed primary school as their highest level of education. Women reported a median household income of RD$10,000–19,999 per month (IQR, RD$5,000–54,999 per month).

151 women (43.1%) were primigravidas, and 199 women (56.9%) were multigravidas. The median gestational age at the time of sample collection was 37 and 1/7 to 38 weeks.

## Medical histories

Sixty-six women (18.9%) reported a chronic medical condition such as asthma or hypertension. Five women (1.4%) were living with human immunodeficiency virus (HIV). Eight women (2.3%) were current smokers, and 49 women (14.0%) had stopped smoking during pregnancy; others had stopped smoking prior to pregnancy or were never smokers. When asked about vaginal douching, 42.9% of women reported practicing vaginal douching, with most (105/150, 70.0%) douching daily. Women reported using a variety of different products for douching including body soap, jabón de cuaba (a household soap common in the DR), and

**Table 1. Demographic characteristics of study participants enrolled at Hospital Materno Infantil San Lorenzo de Los Mina, Santo Domingo, Dominican Republic–total and by rectovaginal Group B *Streptococcus* (GBS) colonization status.**

| | Total | GBS Positive | GBS Negative | p-value |
|---|---|---|---|---|
| | n = 350 | n = 93 | n = 257 | |
| **Country of birth** | | | | 0.47 |
| Dominican Republic | 264 (75.4%) | 73 (78.5%) | 191 (74.3%) | |
| Haiti | 83 (23.7%) | 20 (21.5%) | 63 (24.5%) | |
| Venezuela | 3 (0.9%) | 0 (0%) | 3 (1.2%) | |
| **Maternal age** | | | | 0.98 |
| ≤17 years | 67 (19.1%) | 17 (18.3%) | 50 (19.5%) | |
| 18–24 years | 151 (43.1%) | 42 (45.2%) | 109 (42.4%) | |
| 25–30 years | 83 (23.7%) | 20 (21.5%) | 63 (24.5%) | |
| 31–35 years | 34 (9.7%) | 10 (10.8%) | 24 (9.3%) | |
| 36–40 years | 10 (2.9%) | 3 (3.2%) | 7 (2.7%) | |
| >40 years | 5 (1.4%) | 1 (1.1%) | 4 (1.6%) | |
| **Parity** | | | | 0.48 |
| Primigravida | 151 (43.1%) | 43 (46.2%) | 108 (42.0%) | |
| Multigravida | 199 (56.9%) | 50 (53.8%) | 149 (58.0%) | |
| **Gestational age at enrollment** | | | | 0.28 |
| 35–36 weeks | 61 (17.4%) | 15 (16.1%) | 46 (17.9%) | |
| 36 and 1/7–37 weeks | 75 (21.4%) | 14 (15.1%) | 61 (23.7%) | |
| 37 and 1/7–38 weeks | 88 (25.1%) | 27 (29.0%) | 61 (23.7%) | |
| >38 weeks | 126 (36.0%) | 37 (39.8%) | 89 (34.6%) | |
| **Number of obstetric visits at enrollment** | | | | 0.37 |
| 1 visit | 1 (0.3%) | 0 (0%) | 1 (0.4%) | |
| 2 visits | 0 (0%) | 0 (0%) | 0 (0%) | |
| 3 visits | 6 (1.7%) | 3 (3.2%) | 3 (1.2%) | |
| 4 visits | 5 (1.4%) | 0 (0%) | 5 (1.9%) | |
| 5 visits | 11 (3.1%) | 2 (2.2%) | 9 (3.5%) | |
| >5 visits | 327 (93.4%) | 88 (94.6%) | 239 (93.0%) | |
| **Civil status** | | | | 0.72 |
| Married | 33 (9.4%) | 8 (8.6%) | 25 (9.7%) | |
| Single | 63 (18.0%) | 14 (15.1%) | 49 (19.1%) | |
| Widowed | 1 (0.3%) | 0 (0%) | 1 (0.4%) | |
| Common law marriage | 253 (72.3%) | 71 (76.3%) | 182 (70.8%) | |
| **Education level** | | | | 0.11 |
| Attended some primary school | 184 (52.6%) | 53 (57.0%) | 131 (51.0%) | |
| Primary school | 95 (27.1%) | 20 (21.5%) | 75 (29.2%) | |
| Secondary school | 9 (2.6%) | 5 (5.4%) | 4 (1.6%) | |
| University | 62 (17.7%) | 15 (16.1%) | 47 (18.3%) | |
| **Household income** | | | | 0.76 |
| Less than RD$5000[1] per month | 37 (10.6%) | 9 (9.7%) | 28 (10.9%) | |
| RD$5000–9,999 per month | 92 (26.3%) | 26 (28.0%) | 66 (25.7%) | |
| RD$10,000–19,999 per month | 108 (30.9%) | 27 (29.0%) | 81 (31.5%) | |
| RD$20,000–54,999 per month | 66 (18.9%) | 18 (19.4%) | 48 (18.7%) | |
| RD$55,000 or more per month | 5 (1.4%) | 0 (0%) | 5 (1.9%) | |
| Not known | 42 (12.0%) | 13 (14.0%) | 29 (11.3%) | |

[1]RD$ = Dominican peso.

products specifically marketed for douching. Most women (91.4%) were sexually active during pregnancy, and most (99.4%) reported one sexual partner.

When asked about antimicrobial use, 237 women (67.7%) reported receiving an antimicrobial at least once during pregnancy. Twenty-two women (9.3%) were receiving an antimicrobial at the time of sample collection. Of the women who had received antimicrobials, 32 (13.5%) reported receiving an antibiotic with potential activity against GBS (a beta-lactam, macrolide, or clindamycin). Most women (55.7%) were unable to recall which antimicrobial they had received.

## Obstetric histories

Of the 199 multigravidas, most (42.7%) had one previous pregnancy (Table 2). Sixty-one women (30.7%) reported a history of spontaneous abortion. Sixteen women (8.0%) had given birth prematurely, and six women (3.0%) had experienced stillbirth. Infants born from the previous pregnancies of 17 women (8.5%) required hospitalization for a serious infection before 90 days of age, and infants of three women (1.5%) died. No infant had a known diagnosis of sepsis or meningitis. No infant had known GBS disease. No woman recalled having a rectovaginal culture positive for GBS during a prior pregnancy.

## Prevalence of GBS colonization

Rectovaginal sampling using Carrot Broth identified GBS colonization in 91 women (26.0%). Repeat testing using CHROMagar StrepB identified possible GBS colonization in 107 women

**Table 2. Obstetric histories of 199 multigravid study participants–total and by rectovaginal Group B *Streptococcus* (GBS) colonization status.**

|  | Total | GBS Positive | GBS Negative | p-value |
|---|---|---|---|---|
|  | n = 199 | n = 51 | n = 148 |  |
| **Number of previous pregnancies** |  |  |  | 0.096 |
| 1 | 85 (42.7%) | 24 (47.1%) | 61 (41.2%) |  |
| 2 | 52 (26.1%) | 11 (21.6%) | 41 (27.7%) |  |
| 3 | 33 (16.6%) | 4 (7.8%) | 29 (19.6%) |  |
| 4 | 21 (10.6%) | 9 (17.6%) | 12 (8.1%) |  |
| ≥5 | 8 (4.0%) | 3 (5.9%) | 5 (3.4%) |  |
| **Number of living children** |  |  |  | 0.057 |
| 0 | 29 (14.6%) | 8 (15.7%) | 21 (14.2%) |  |
| 1 | 80 (40.2%) | 22 (43.1%) | 58 (39.2%) |  |
| 2 | 53 (26.6%) | 7 (13.7%) | 46 (31.1%) |  |
| 3 | 23 (11.6%) | 7 (13.7%) | 16 (10.8%) |  |
| 4 | 7 (3.5%) | 4 (7.8%) | 3 (2.0%) |  |
| 5 | 2 (1.0%) | 0 (0%) | 2 (1.4%) |  |
| ≥6 | 5 (2.5%) | 3 (5.9%) | 2 (1.3%) |  |
| **Outcomes of prior pregnancies** |  |  |  |  |
| Spontaneous abortion | 61 (30.7%) | 16 (31.4%) | 45 (30.4%) | 0.90 |
| Premature birth | 16 (8.0%) | 5 (9.8%) | 11 (7.4%) | 0.59 |
| Stillbirth | 6 (3.0%) | 2 (3.9%) | 4 (2.7%) | 0.66 |
| **Infant(s) with a serious infection requiring hospitalization at <90 days of age** |  |  |  |  |
| Sepsis | 0 (0%) | 0 (0%) | 0 (0%) | — |
| Meningitis | 0 (0%) | 0 (0%) | 0 (0%) | — |
| Pneumonia | 4 (2.0%) | 1 (2.0%) | 3 (2.0%) | 0.91 |
| Other infection | 13 (6.5%) | 6 (11.8%) | 7 (4.7%) | 0.18 |
| Death | 3 (1.5%) | 1 (2.0%) | 2 (1.4%) | 0.87 |

(30.6%). Confirmatory sequencing identified GBS in 93 (26.6%) of the 107 samples that were positive on CHROMagar. Of these, twelve samples had negative Carrot Broth results. Fourteen samples that had colony growth consistent with GBS on CHROMagar were determined by sequencing to be non-GBS species, including *Lactococcus lactis*, *Streptococcus sanguinis*, and *Aerococcus* species, which have been reported to mimic GBS on chromogenic media [23]. When compared to molecular methods in this cohort, Carrot Broth was therefore 87.1% sensitive and 96.1% specific for detecting GBS (Kappa = 0.84 [95% CI, 0.73–0.94]).

Six of the ten known GBS serotypes were found (Fig 1). Serotype III was identified most frequently, in 21.5% of isolates, followed by type II (19.4%), type IV (18.3%), type Ia (16.1%), type V (15.1%), and type Ib (9.7%). Serotypes VI, VII, VIII, and IX were not detected. The most

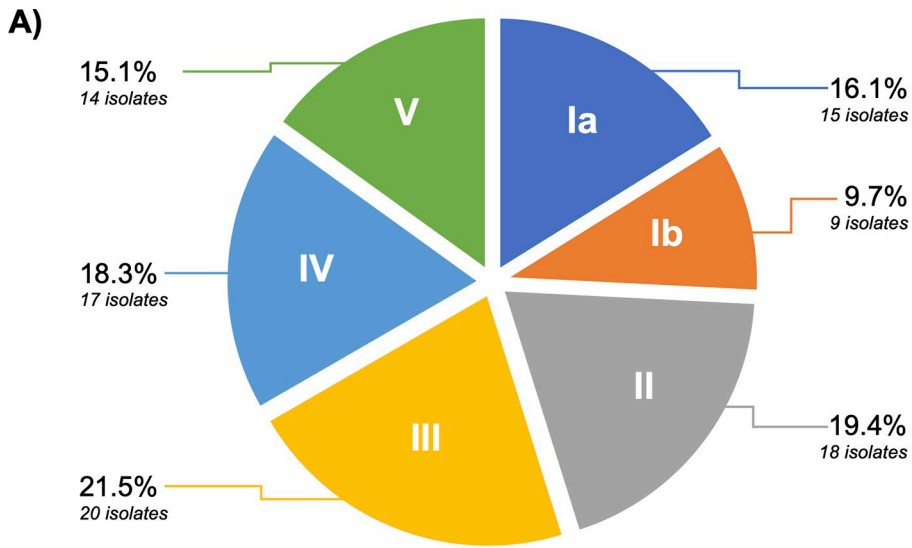

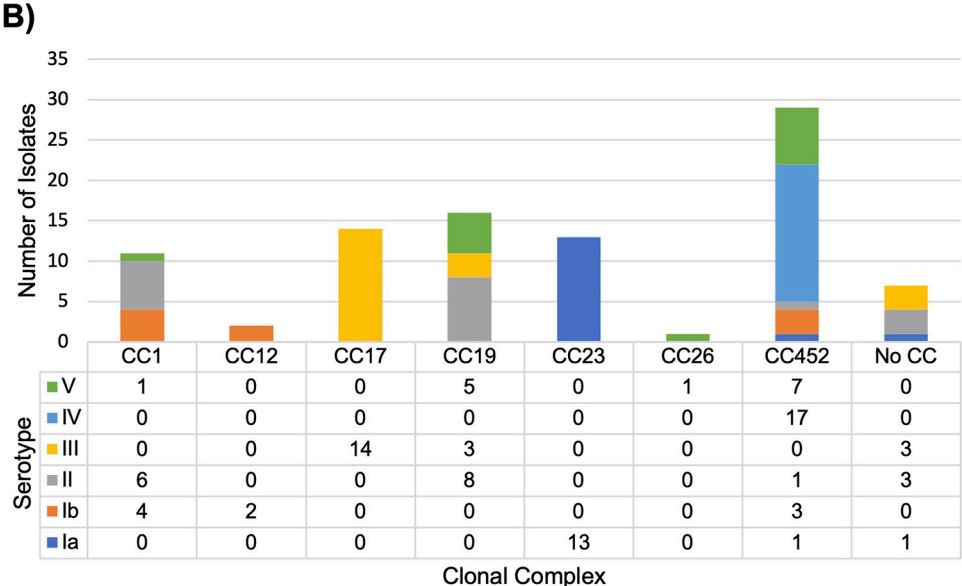

**Fig 1. Characteristics of 93 Group B *Streptococcus* isolates identified in rectovaginal samples from near-term pregnant women in the Dominican Republic.** A) Serotypes. B) Clonal complexes.

common sequence types were ST1010 (17.2%) and ST17 (14.0%). When characterized by clonal complex, the most common were CC452 (30.1%), CC19 (18.3%), CC17 (15.1%), and CC23 (15.1%). The most common pairings of serotypes and MLST were IV/ST1010 (17.2%), III/CC17 (14.0%), and Ia/CC23 (9.7%). Four isolates had allele combinations that have not been previously described, representing novel MLSTs. These were assigned new MLST designations (STs 2113–2116) by PubMLST. Isolate-level MLST, CC, serotype, and genomic prediction of antimicrobial resistance are presented in S1 Table. No isolates had PBP2x mutations known to confer reduced susceptibility to beta-lactam antibiotics. Predicted resistance to tetracycline was common, occurring in 72/93 (77.4%) of strains. MLS resistance determinants were predicted in 26/93 (27.9%), and one strain had a first-step mutation in the *parC* gene that is associated with the development of resistance to fluoroquinolones. A tree based on genetic distances among isolates showed overall concordance with CC assignments (Fig 2).

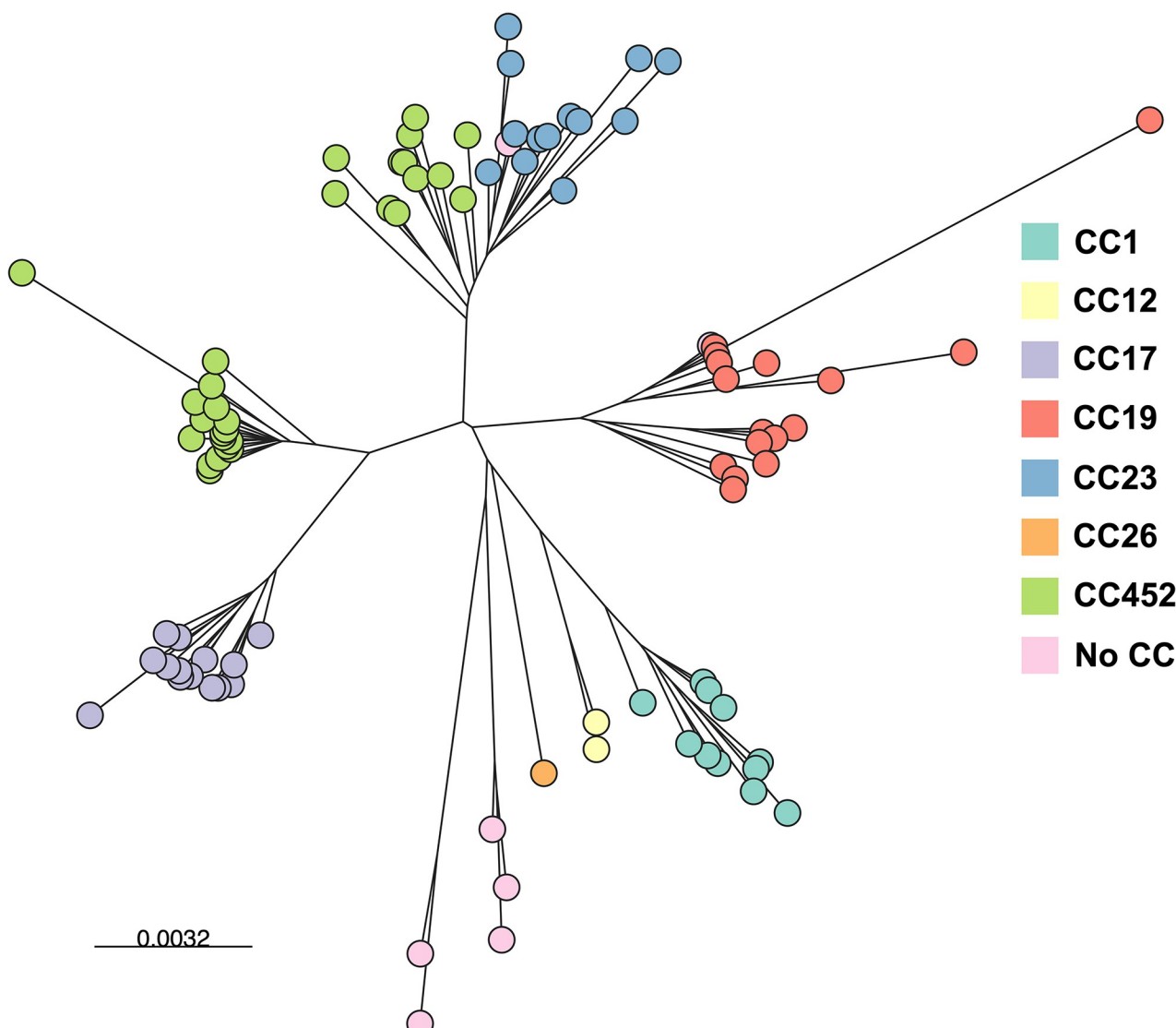

**Fig 2. Genomic diversity of 93 Group B *Streptococcus* (GBS) isolates identified in rectovaginal samples from near-term pregnant women in the Dominican Republic.** A kmer-based tree of GBS whole-genome sequences was constructed using Mashtree and visualized in Microreact. Nodes are color coded by clonal complex as indicated. Branch lengths and the scale bar represent Mash distances.

### Risk factors for GBS colonization

A secondary analysis found no statistically significant association between any demographic, medical history, or obstetric history variable and the presence of rectovaginal GBS colonization, when detected using WGS (Tables 1 and 2). There was also no significant association with the various serotypes and clonal complexes. Similarly, no statistically significant association was found between these variables and the presence of rectovaginal GBS colonization when detected using either Carrot Broth or CHROMagar.

## Discussion

We identified rectovaginal GBS colonization in 26.6% of near-term pregnant women who presented for routine prenatal care at an urban tertiary referral center in the DR. This high prevalence is similar to previous pooled estimates from the greater Caribbean region and is nearly 50% greater than the estimated global prevalence of maternal GBS colonization [2].

This is the first report on GBS serotypes identified in maternal colonization in the DR. In this cohort, we only observed GBS serotypes Ia, Ib, II, III, IV, and V. No one serotype predominated. By contrast, global analyses of maternal GBS colonization have shown an uneven distribution of serotypes, with serotypes III and Ia most prevalent overall [10]. In Latin America and the Caribbean, serotype III has been identified less often, and serotypes Ia and II have been most prevalent [10]. The distribution of GBS serotypes colonizing pregnant women elsewhere in the Caribbean region has only previously been described in Trinidad and Tobago, where serotypes II and Ib predominated (47.2% and 23.6%, respectively) [24]. In this cohort, serotypes III and II were most prevalent (21.5% and 19.4%, respectively). We also observed an unusually high prevalence of serotype IV (18.3%). Serotype IV occurs rarely across all regions but has been identified more frequently in recent studies, perhaps as a result of genomic recombination between different GBS CCs [10, 24]. Strikingly, all seventeen serotype IV isolates in our cohort were identified as ST1010, suggesting local expansion of this GBS strain in the DR. ST1010 has not been widely reported in other settings, and its characteristics in clinical disease are unknown [25]. In an analysis of 6340 invasive GBS strains from the United States, only 7 (0.1%) were ST1010 [25].

A high proportion of GBS isolates in this cohort belonged to CCs that have been associated with increased risk for neonatal or maternal invasive disease. We identified fourteen isolates (15.1%) belonging to the hypervirulent CC17; all isolates were serotype III. CC17 strains cause most episodes of neonatal late-onset invasive GBS disease worldwide [10, 26]. CC452 was the most prevalent CC identified, representing 30.1% of all isolates. A recently recognized entity, CC452 likely emerged through genomic recombination between CC23 and CC17 lineages, and strains have been described in Canada and the United States as causes of neonatal invasive disease [27–29]. Other frequently identified CCs in this cohort included CC19 (18.3%), which has been associated with early-onset invasive GBS disease and increased mortality in neonates, and CC23 (15.1%), which has been associated with maternal invasive disease [10, 26, 30, 31]. We identified four novel GBS MLST profiles in this cohort, three of which were serotype II strains. These results–a combination of known, emerging, and novel strains–underscore the ongoing critical importance of WGS to better understanding the global distribution and evolution of GBS strains.

Our findings illustrate the diagnostic challenges in detecting rectovaginal GBS colonization. There is no agreed-upon standard laboratory method for identifying GBS in rectovaginal samples. In addition, commonly available methods have limitations that may affect sensitivity or specificity. Carrot Broth does not detect non-hemolytic GBS strains; for this reason, the manufacturer recommends using culture-based methods to confirm all samples with negative Carrot Broth results. Conversely, CHROMagar StrepB may yield falsely positive results in the presence of some non-GBS species that may colonize humans.

In this cohort, Carrot Broth detected rectovaginal GBS colonization with generally good sensitivity and specificity when compared to higher-cost, more resource-intensive culture-based or molecular methods. Rectovaginal swabs can be transferred directly to Carrot Broth without the need for additional media, and the chromogenic change that occurs in the presence of hemolytic GBS strains is detectable by eye. Carrot Broth-based screening thus offers a potential approach for expanding maternal screening strategies in resource-limited settings. We did encounter logistical challenges that may impact the feasibility of Carrot Broth-based screening in settings with inconsistent access to refrigeration and stable supply chains: the medium has a short shelf life, and it must be stored at 2–8°C and protected from light to prevent deterioration.

Previous studies of maternal rectovaginal GBS colonization have identified potential associations with a range of maternal factors including age, socioeconomic status, pre-pregnancy body mass index, tobacco use, antimicrobial exposure, vaginal douching practices, and prior GBS colonization [2, 32, 33]. In this cohort, we found no statistically significant association between these variables and the presence of rectovaginal GBS colonization. However, our sample was likely underpowered to detect such differences. Recall bias may have also affected assessment of variables such as pre-pregnancy BMI and antimicrobial exposure. Larger studies are needed to better determine which factors contribute to rectovaginal GBS colonization, particularly in regions with a high prevalence of colonization.

These results show that pregnant women in the DR experience high rates of rectovaginal colonization with virulent GBS strains. While it was beyond the scope of this study to include neonatal follow-up, these results suggest that rates of neonatal GBS infection are likely to be high in this region. Rivera and colleagues previously found a very high incidence of early onset GBS disease among infants at two sites in the DR, affecting 2.34 infants per 1000 live births; no mothers of affected infants had received routine GBS screening prior to delivery, and only 17% of mothers had received risk-based IAP. Case fatality rate in this study was as high as 40.0% (95% CI, 19.8–64.3%) [6]. In contrast, the incidence of early onset GBS disease in infants in Latin America and the Caribbean overall has been measured at 0.51 per 1000 live births (95% CI, 0.3–0.7%), though more recent estimates suggest that this undercalculates the true burden of neonatal disease in the region [1, 8]. The hypervirulent GBS strains observed in our cohort of colonized pregnant women may contribute to some of the high disease incidence seen in neonates in the DR. In clinical practice, limited access to laboratory materials in the DR creates challenges for identifying GBS in neonates, and many ill neonates receive treatment for presumed invasive GBS disease without culture-based confirmation. Hexavalent GBS vaccines now in development target all of the serotypes that we observed, suggesting that these candidate vaccines could offer effective protection against maternal and neonatal GBS disease in the DR [7, 11].

Limitations of this study include 1) single center focus, 2) inclusion only of pregnant women seeking antenatal care, 3) small sample size for detecting associations between maternal GBS colonization and demographic variables, 4) reliance on participants' recall to record medical history details, and 5) inability to generalize results to pregnant women in the broader Caribbean region. Nevertheless, we identified a high prevalence of rectovaginal GBS colonization in near-term pregnant women in the DR and provided the first description of those GBS strains. Future studies will need to better characterize the incidence, causative strains, and outcomes of neonatal invasive GBS disease in the DR.

## Supporting information

**S1 Checklist. Inclusivity in global research.**
(DOCX)

**S2 Checklist. STROBE statement.**
(DOCX)

**S1 Fig. Pregnant women assessed for study eligibility.**
(DOCX)

**S1 Table. Isolate-level data for Group B *Streptococcus* (GBS) strains collected in late pregnancy in the Dominican Republic.**
(DOCX)

## Acknowledgments

We acknowledge the NYU Langone Antimicrobial-Resistant Pathogens Program (Dr. Victor Torres and Dr. Bo Shopsin) and the NYU Langone Health Genome Technology Center for assistance with GBS whole-genome sequencing. We also acknowledge Adriana Deverlis, formerly of the CHOP Global Health Center, who helped lay the groundwork for this study.

## Author Contributions

**Conceptualization:** Sandra Valera, Anabel Fernández, Andrew P. Steenhoff, Adam J. Ratner, Sarah Geoghegan.

**Data curation:** Katherine M. Laycock, Francia Acosta, Rosa Felipe, Anabel Fernández, Megan Job, Sophia Dongas, Adam J. Ratner, Sarah Geoghegan.

**Formal analysis:** Katherine M. Laycock, Megan Job, Adam J. Ratner.

**Funding acquisition:** Sandra Valera, Anabel Fernández, Andrew P. Steenhoff, Adam J. Ratner, Sarah Geoghegan.

**Investigation:** Katherine M. Laycock, Francia Acosta, Rosa Felipe, Anabel Fernández, Megan Job, Sophia Dongas, Adam J. Ratner, Sarah Geoghegan.

**Methodology:** Sandra Valera, Anabel Fernández, Andrew P. Steenhoff, Adam J. Ratner, Sarah Geoghegan.

**Project administration:** Francia Acosta, Sandra Valera, Ana Villegas, Elia Mejia, Christian Mateo, Rosa Felipe, Anabel Fernández, Andrew P. Steenhoff, Sarah Geoghegan.

**Validation:** Katherine M. Laycock, Francia Acosta, Rosa Felipe, Megan Job, Sophia Dongas, Andrew P. Steenhoff, Adam J. Ratner, Sarah Geoghegan.

**Visualization:** Katherine M. Laycock.

**Writing – original draft:** Katherine M. Laycock.

**Writing – review & editing:** Katherine M. Laycock, Francia Acosta, Sandra Valera, Ana Villegas, Elia Mejia, Christian Mateo, Rosa Felipe, Anabel Fernández, Megan Job, Sophia Dongas, Andrew P. Steenhoff, Adam J. Ratner, Sarah Geoghegan.

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
