## [Decision Letter · Decision Letter 0]

10 Jul 2023

PGPH-D-23-00911

Near-term pregnant women in the Dominican Republic experience high rates of Group B Streptococcus rectovaginal colonization with virulent strains

Dear Dr. Laycock,

Thank you for submitting your manuscript to PLOS Global Public Health. After careful consideration, we feel that it has merit but does not fully meet PLOS Global Public Health’s publication criteria as it currently stands. Therefore, we invite you to submit a revised version of the manuscript that addresses the points raised during the review process.

Based on the positive feedback from the reviewers, we would like to further consider this paper for publication. Therefore, we would like you to revise the manuscript based on the reviewers' feedback below. Please pay special attention to the comments regarding the statistical and genomic analysis.

We look forward to receiving your revised manuscript.

Kind regards,

Chrispin Chaguza, Ph.D

Academic Editor

Journal Requirements:

2. We noticed that you used "unpublished data" in the manuscript. We do not allow these references, as the PLOS data access policy requires that all data be either published with the manuscript or made available in a publicly accessible database. Please amend the supplementary material to include the referenced data or remove the references.

Additional Editor Comments (if provided):

I suggest including antimicrobial resistance profiles based on the generated whole-genome sequencing data. Since the genomic data is already available, you could also optionally include a phylogenetic tree and describe the genomic diversity of the GBS isolates beyond the serotypes and MLST, for example, looking at the number of mutational (single nucleotide polymorphisms [SNP]) differences between isolates of the same ST/CC.

Reviewers' comments:

Reviewer's Responses to Questions

**Comments to the Author**

1. Does this manuscript meet PLOS Global Public Health’s publication criteria? Is the manuscript technically sound, and do the data support the conclusions? The manuscript must describe methodologically and ethically rigorous research with conclusions that are appropriately drawn based on the data presented.

Reviewer #1: Yes

Reviewer #2: Yes

2. Has the statistical analysis been performed appropriately and rigorously?

Reviewer #1: No

Reviewer #2: Yes

3. Have the authors made all data underlying the findings in their manuscript fully available (please refer to the Data Availability Statement at the start of the manuscript PDF file)?

Reviewer #1: Yes

Reviewer #2: Yes

4. Is the manuscript presented in an intelligible fashion and written in standard English?

Reviewer #1: Yes

Reviewer #2: Yes

5. Review Comments to the Author

Reviewer #1: In this manuscript, the authors seek to describe GBS prevalence among mothers seeking antenatal care at a healthcare facility in the Dominican Republic. They also describe the distribution of GBS clones circulating among this population to gauge the impact that GBS vaccines may have in preventing disease. This is a clear, concisely written, and important manuscript. I have only minor comments.

Line 127 – “Age bands” – please define this term.

Line 154 – What do you mean by “contamination checks”? What threshold was used to determine contamination – for example, did you require that 100% of reads be classified as GBS to classify an isolate as “pure”?

Line 162 – What is expected level of confidence and/or precision with a sample size of 350?

Line 167: Please indicate in parentheses what methods of GBS detection were used, to remind readers.

Line 169: Why didn’t you use t-tests to examine associations between continuous exposures and GBS prevalence, for example age, # of previous pregnancies, etc in Tables 1 and 2? You classification of some continuous variables into categories seems arbitrary for the purposes of statistical analysis.

Line 171: Mention p-value used to determine statistical significance.

Line 241: “Four isolates had allele combinations that have not been previously described.” Are you referring to MLSTs here? If so, please state this. Also, you should assign STs for novel ST types and include these designations in parentheses here. This should be straightforward if you have WGS data.

Line 268-270: Text appears to contrast what is reported in line 262, where you write that only GBS serotypes Ia, Ib, and II-V were observed in your cohort. How then are “serotypes III and II

… most prevalent” as reported in line 268?

Line 322 – Please qualify “very high.” What is the typical rate in other Latin American countries with more resources and/or greater access to maternal care?

Reviewer #2: The article meets the criteria for PLOS Global public health journal publication and has a technically sound manuscript with conclusions that are supported by data presented in the results section. The mixed methods design for the primary research question posed undoubtedly projects an ethically and methodologically sound design with mentioned limitations and weaknesses.

This is further elaborated by the drawn conclusions which even with the mentioned biases in the design, not only answer the question posed but go on to identify the significance of the the hexavalent vaccine against the local group B streptococcus serotypes in the Dominican Republic. Despite the mentioned practical hindrances to implementing such a vital public health intervention in any developing nation, the conclusions also justify the development of better vaccines with more serotypes coverage through the recognition of 4 new and probably virulent group B streptococcus serotypes in a nation that might struggle to screen and prevent group b infections on top of the poor performances of the streptococcus isolation and diagnostic technology (carrot broth and Chromoagar).

An astounding attempt at statistical analysis of the data is seen through the sub-categorization of data captured from the interviews and attempt to find associations and correlations between the data but the mathematical formulas that were used to infer or defer any data correlation or association was just mentioned and not fully described. However, data availability seems adequate through data projected in the tables and the manuscript is presented in good palatable english.

A recommendation would be to provide the identification of 4 unrecognized species earlier in the results paragraph before mentioning the calculated sensitivity and specificity of the carrot-broth method of identifying and isolating the group b streptococcus. This is because of the logical presentation of the data in that paragraph which provides the numerical data to calculate sensitivity and specificity of the carrot-broth technique, but one would assume the provided figures, 87.1% and 96% to be wrong if you are working with a total of 350 eligible women who were tested yet 4 women had new strains that could not be classified such that the total tests included in the analysis became 346 instead. Otherwise this is an insightful research article that is relevant for its purpose and meets publication criteria.

6. PLOS authors have the option to publish the peer review history of their article (what does this mean?). If published, this will include your full peer review and any attached files.

**Do you want your identity to be public for this peer review?** For information about this choice, including consent withdrawal, please see our Privacy Policy.

Reviewer #1: No

Reviewer #2: **Yes: **Benson Tarisai Gombe

---

## [Decision Letter · Decision Letter 1]

22 Aug 2023

Near-term pregnant women in the Dominican Republic experience high rates of Group B Streptococcus rectovaginal colonization with virulent strains

PGPH-D-23-00911R1

Dear Dr. Laycock,

We are pleased to inform you that your manuscript 'Near-term pregnant women in the Dominican Republic experience high rates of Group B Streptococcus rectovaginal colonization with virulent strains' has been provisionally accepted for publication in PLOS Global Public Health.

Best regards,

Chrispin Chaguza, Ph.D

Academic Editor

Reviewer Comments (if any, and for reference):

Reviewer's Responses to Questions

**Comments to the Author**

1. If the authors have adequately addressed your comments raised in a previous round of review and you feel that this manuscript is now acceptable for publication, you may indicate that here to bypass the “Comments to the Author” section, enter your conflict of interest statement in the “Confidential to Editor” section, and submit your "Accept" recommendation.

Reviewer #1: All comments have been addressed

Reviewer #2: All comments have been addressed

2. Does this manuscript meet PLOS Global Public Health’s publication criteria? Is the manuscript technically sound, and do the data support the conclusions? The manuscript must describe methodologically and ethically rigorous research with conclusions that are appropriately drawn based on the data presented.

Reviewer #1: Yes

Reviewer #2: Yes

3. Has the statistical analysis been performed appropriately and rigorously?

Reviewer #1: Yes

Reviewer #2: Yes

4. Have the authors made all data underlying the findings in their manuscript fully available (please refer to the Data Availability Statement at the start of the manuscript PDF file)?

Reviewer #1: Yes

Reviewer #2: Yes

5. Is the manuscript presented in an intelligible fashion and written in standard English?

Reviewer #1: Yes

Reviewer #2: Yes

6. Review Comments to the Author

Reviewer #1: (No Response)

Reviewer #2: The article meets the criteria for PLOS Global public health journal publication and has a technically sound manuscript with conclusions that are supported by data presented in the results section. The mixed methods design for the primary research question posed projects an ethically and methodologically sound design with mentioned limitations and weaknesses. This is further elaborated by the drawn conclusions which even with the mentioned biases in the design, not only answer the question posed but go on to identify the significance of the the hexavalent vaccine against the local group B streptococcus serotypes in the Dominican Republic.

Despite the mentioned practical hindrances to implementing such a vital public health intervention in any developing nation, the conclusions also justify the development of better vaccines with more serotypes coverage through the recognition of 4 new and probably virulent group B streptococcus serotypes in a nation that might struggle to screen and prevent group b infections .

An astounding attempt at statistical analysis of the data is seen through the sub-categorization of data captured from the interviews and attempt to find associations and correlations between the data but the mathematical formulas that were used to infer or defer any data correlation or association was not fully described . However, data availability seems adequate through data projected in the tables. The manuscript is presented in good palatable english.

7. PLOS authors have the option to publish the peer review history of their article (what does this mean?). If published, this will include your full peer review and any attached files.

**Do you want your identity to be public for this peer review?** For information about this choice, including consent withdrawal, please see our Privacy Policy.

Reviewer #1: No

Reviewer #2: No
